# Differentiated Thyroid Cancer Long-Term Outcomes and Risk Stratification in Pediatric and Adolescent Patients: A 44-Year Retrospective Study

**DOI:** 10.3390/diagnostics15040399

**Published:** 2025-02-07

**Authors:** Marko Buta, Nada Santrac, Merima Goran, Nikola Jeftic, Jovan Raketic, Nevena Savkovic, Saska Pavlovic, Milan Zegarac, Neven Jokic, Ana Cvetkovic, Marko Stojanovic, Ana Jotic, Dusica Gavrilovic, Ivan Markovic

**Affiliations:** 1School of Medicine, University of Belgrade, Dr Subotica 8, 11 000 Belgrade, Serbia; markobuta@gmail.com (M.B.); santrac.nada@gmail.com (N.S.); merimaoruci@hotmail.com (M.G.); milan_zegarac@yahoo.com (M.Z.); anadjurdjic@yahoo.com (A.C.); markostoj@yahoo.com (M.S.); anajotic@yahoo.com (A.J.); ivanmarkovic66@yahoo.com (I.M.); 2Surgical Oncology Clinic, Institute for Oncology and Radiology of Serbia, Pasterova 14, 11 000 Belgrade, Serbia; jefta20@gmail.com (N.J.); nevena5gm@gmail.com (N.S.); saskapavlovic23@gmail.com (S.P.); anva.neven@gmail.com (N.J.); 3Department of Anesthesiology, Institute for Oncology and Radiology of Serbia, Pasterova 14, 11 000 Belgrade, Serbia; 4Clinic for Endocrinology, Diabetes and Metabolic Diseases, University Clinical Center of Serbia, Dr Subotica 13, 11 000 Belgrade, Serbia; 5Clinic for Otorhinolaryngology and Maxillofacial Surgery, University Clinical Center of Serbia, Pasterova 2, 11 000 Belgrade, Serbia; 6Data Center, Institute of Oncology and Radiology of Serbia, Pasterova 14, 11 000 Belgrade, Serbia; duca.gavrilovic@gmail.com

**Keywords:** differentiated thyroid cancer, children, adolescents, prognostic factors, long-term outcomes, risk stratification

## Abstract

**Background:** Differentiated thyroid cancer (DTC) in children and adolescents is a rare but significant malignancy, often presenting at more advanced stages compared to adults, although it is associated with favorable long-term outcomes. This study aimed to identify prognostic factors and perform risk stratification with the goal of identifying low-risk patients who would benefit from a less radical treatment approach. **Methods:** This retrospective cohort study included patients aged 21 years and younger with DTC treated at the Institute for Oncology and Radiology of Serbia between 1980 and 2024. **Results:** The study analyzed 99 patients (39 children, 60 adolescents) with a median follow-up of 15.6 years (range: 0.6–43.6 years). No significant differences in long-term outcomes were observed between children and adolescents. Multivariate regression analysis identified a total number of more than 10.5 positive lymph nodes and extrathyroidal tumor extension as independent predictors of adverse events and event-free interval (EFI). Using these prognostic factors, patients were stratified into three groups: low-risk (no risk factors), intermediate-risk (one of two risk factors), and high-risk (both risk factors). Statistically significant differences in EFI were observed among the three groups. Notably, none of the patients in the low-risk group had evidence of disease after treatment. Patients classified as having no evidence of disease after treatment demonstrated significantly better EFI compared to those with evidence of disease. **Conclusions:** Our findings highlight the importance of meticulous risk stratification in predicting long-term outcomes and might serve as a basis for developing personalized therapeutic strategies. Identifying low-risk patients who may benefit from a less aggressive treatment approach while ensuring optimal treatment and follow-up for high-risk patients remains a central objective in the modern management of DTC.

## 1. Introduction

Thyroid cancer is a rare yet significant malignancy in children and adolescents, with differentiated thyroid cancer (DTC) being the most prevalent. Worldwide, the annual incidence of DTC in children ranges from 0.5 to 10.0 cases per 100,000 [1]. According to the Institute of Public Health of Serbia “Dr Milan Jovanovic Batut”, the incidence of DTC in individuals under 20 years of age was reported to be 0.37 cases per 100,000 in 2018 [2]. In prepubertal children, DTC occurs at similar rates in males and females. However, with the onset of puberty, the female-to-male ratio increases to as high as 6:1, making thyroid cancer the second most common malignancy in adolescent girls [3]. The incidence of DTC is rising globally among children, adolescents, and young adults. While improved detection efforts contribute to this trend, they alone cannot fully explain the rise. Potential environmental, dietary, and genetic factors should be explored to better understand the underlying causes [4]. DTC is frequently diagnosed as a radiation-induced second primary malignancy (SPM), particularly in survivors of childhood lymphoma and leukemia who have undergone radiotherapy [5,6].

Thyroid nodules in children warrant thorough evaluation due to their significantly higher malignancy rates compared to adults, ranging from 10% to 50% [7]. Additionally, children with DTC often present with more advanced disease, characterized by a higher incidence of lymph node involvement at diagnosis (up to 90% in children versus 20–50% in adults) and a greater frequency of distant metastases, particularly to the lungs (20–30% in children compared to less than 10% in adults) [8,9,10,11]. Despite pediatric and adolescent patients often presenting with more advanced disease, their prognosis is generally more favorable than that of adults [1]. Achieving a cure while minimizing treatment-associated morbidity has become the primary objective in the modern management of DTC in children and adolescents [8]. In 2015, the American Thyroid Association (ATA) released its first guidelines for the treatment of well-differentiated thyroid cancer in children, which have since become a cornerstone of clinical practice for managing this condition [12]. The most recent guidelines for pediatric DTC were published by the European Thyroid Association (ETA) in 2022 [13]. The development of specific guidelines for the pediatric population underscores the recognition that this disease represents a distinct entity in children, differing significantly from its presentation and management in adults.

This retrospective study aimed to identify prognostic factors of DTC in children and adolescents and perform risk stratification with the goal of identifying low-risk patients who would benefit from a less radical treatment approach.

## 2. Materials and Methods

### 2.1. Patients and Data

This retrospective cohort study was conducted on patients treated between 1980 and 2024 at the Institute for Oncology and Radiology of Serbia (IORS), including those who had a part of their treatment outside IORS. IORS is a tertiary healthcare center housing both a Surgical Oncology Clinic and a Pediatric Oncology Clinic. It is one of the two largest institutions in Serbia specializing in thyroid surgery, managing an average of over 750 patients annually, with a total of 33,659 patients treated over a 44-year period. This study included all patients aged 21 years and younger with DTC. Patients with other types of thyroid cancers and with incomplete medical records were excluded from the study. As a part of the preoperative evaluation, all patients underwent neck and abdominal ultrasound, tracheal and chest X-ray examinations, laryngoscopy for vocal cord function evaluation, thyroid function analysis, and anesthesiologic assessment of perioperative risk. The final treatment strategy was determined by a multidisciplinary team (tumor board for endocrine pathology). Preoperative, operative, and follow-up data were extracted from the clinic’s patient database. Any missing information was retrospectively obtained from patient records or via telephone inquiries. Following comprehensive medical record revision, all patients were restaged according to the pathological Tumor-Node-Metastasis (pTNM) classification of the AJCC/UICC 8th edition to ensure data uniformity [14]. Data was validated by two independent researchers. This study received approval from the IORS Ethics Committee.

### 2.2. Treatment

The treatment plan for patients with DTC at IORS included total (or near total) thyroidectomy combined with central neck dissection. Based on institutional protocols, patients without clinically evident lateral neck metastases underwent sentinel lymph node biopsy using an original technique by Professor Radan Dzodic, previously published in referent literature [15]. If lateral neck metastases were clinically evident or if the sentinel lymph node biopsy result was positive, modified radical neck dissection of the affected side was conducted. All patients were operated on by high-volume thyroid surgeons, each performing more than 50 thyroid surgeries annually. Tumor size, number of positive lymph nodes, capsular invasion, multifocality, and extrathyroidal extension were reported after a thorough histopathological examination conducted by experienced pathologists. An extrathyroidal extension was defined as the spread of thyroid cancer beyond the thyroid gland into surrounding tissues. This definition encompassed both minimal and gross extrathyroidal extension, as the specific subtypes were not routinely reported for a significant number of patients. Following surgical treatment, patients with multifocal or multicentric tumors, locoregionally advanced disease, or radioiodine-avid metastatic disease received radioactive iodine (RAI) treatment. Patients with locally advanced unresectable disease or non-radioiodine-avid metastatic disease received external beam radiotherapy. All patients were prescribed lifelong postoperative TSH-suppressive therapy with L-thyroxine. Patients with negative post-RAI treatment whole-body 131I scan and those with negative neck ultrasound, if they did not undergo RAI treatment, were classified as having no evidence of disease (NED). Patients with residual disease after treatment were classified as having evidence of disease (ED).

### 2.3. Follow-Up and Long-Term Outcomes

The follow-up regimen included quarterly appointments during the first year post-surgery, shifting to 6-month visits from the second through the fifth year, and annual check-ups thereafter. At each visit, patients underwent a comprehensive physical examination with neck ultrasound and laboratory analyses, including TSH, FT4, thyroglobulin, and thyroglobulin antibodies. As there were no deaths, long-term outcomes were evaluated through disease-free interval (DFI), progression-free interval (PFI), and event-free interval (EFI). DFI was calculated only for patients classified as NED after treatment, while PFI was calculated only for patients classified as ED. EFI was calculated for the entire study population. DFI and PFI refer to the period from completion of initial treatment to disease relapse and progression, respectively. EFI was defined as the time from initial treatment completion to any adverse event (local or distant disease relapse or disease progression). For all long-term outcomes, the last check-up was used as a censoring date for living patients without disease relapse.

### 2.4. Statistical Analysis

Categorical variables were described using frequencies (percentages), while mean, median, standard deviation (SD), and range were used for numeric variables. Firstly, patients were stratified into two groups: children aged 16 years and younger and adolescents aged 17 years and older to examine potential differences between these two age groups. Besides age, the entire study population was tested to identify potential prognostic factors for adverse event occurrence. Receiver operating characteristic (ROC) curve analysis was conducted to assess the accuracy of tumor size and the total number of positive lymph nodes in predicting the occurrence of adverse events. The area under the ROC curve (AUC ROC) was calculated using DeLong’s method, and the likelihood ratio test was applied to evaluate the AUC ROC. The optimal cutoff values for tumor size and the total number of positive lymph nodes were determined as the points with maximum sensitivity and specificity. Patients were further categorized based on these cutoff values, provided the AUC ROC exceeded 70% [16]. The predictive value of all variables significantly associated with adverse events was assessed by univariate and a multivariate logistic regression model. Results were given as an odds ratio with a 95% confidence interval. Survival analysis methods were used to analyze long-term treatment outcomes (Kaplan–Meier method for cumulative probability curves; median with 95% confidence interval-CI; log-rank test). Variables identified as significant by survival analysis were further evaluated using univariate and multivariate Cox proportional hazards regression models (Hazard ratio-HR with 95% CI; Wald and Likelihood ratio tests). Significant prognostic factors were subsequently utilized to stratify the study population into three risk categories: low-risk, intermediate-risk, and high-risk. A *p*-value of less than 0.05 was deemed statistically significant. Statistical analyses were performed using R software (version 4.3.1 (2023-06-16 ucrt)—“Beagle Scouts”; Copyright (C) 2023 The R Foundation for Statistical Computing; Platform: x86_64-w64-mingw32/x64 (64-bit)) (available at: www.r-project.org; downloaded: 21 August 2023).

## 3. Results

### 3.1. Patients and Treatment

A total of 99 patients met the inclusion criteria for this study, with a median follow-up of 15.6 years (range: 0.6–43.6 years). The demographical and treatment characteristics of the study cohort are summarized in Table 1. Of these, 39 (39.39%) were children aged 16 years or younger, while the remaining 60 (60.61%) were adolescents. Female patients predominated in both groups. Four patients had a history of prior radiation therapy for cancer treatment: three received whole-body irradiation for acute lymphoblastic leukemia, and one underwent neck irradiation for Hodgkin lymphoma. At diagnosis, 85 (85.86%) patients presented with clinical signs or symptoms, with thyroid nodules, neck lymphadenopathy, or both observed in 79.8% of cases. The remaining patients were asymptomatic at presentation.

Fine-needle aspiration biopsy (FNAB) was performed in 23 (23.23%) patients, while others were operated on due to suspicious clinical findings. Nineteen (19.19%) patients presented with cervical lymphadenopathy and were diagnosed following surgical removal of pathological cervical lymph nodes. Eight (8.08%) patients had distant metastatic disease at the time of diagnosis, all with lung involvement. All patients underwent surgical treatment. Among them, 75 (75.75%) received radioactive iodine (RAI) treatment and four required postoperative external beam radiotherapy.

### 3.2. Postoperative Staging, Pathohistological Characteristics, and Long-Term Outcomes

As shown in Table 2, the most common histological type in the cohort was papillary thyroid cancer (PTC), identified in 95 (95.96%) patients. Children had a slightly larger mean tumor size compared to adolescents, though this difference was not statistically significant. No significant difference was noted in the number of positive central and lateral lymph nodes between the two groups. On the other hand, in the subgroup of patients that had at least one positive lymph node, children had a significantly higher number of positive lateral lymph nodes compared to adolescents with 9.38 (SD 6.70) vs. 5.95 (SD 5.76) positive lymph nodes; *p* = 0.02. Multifocal tumors, capsular invasion, and extrathyroidal extension were more frequently observed in children than in adolescents. A total of 91 (91.91%) patients were classified as NED after completing treatment, and there was no significant difference between children and adolescents in the proportion of NED and ED patients.

Long-term treatment outcomes are summarized in Table 3. In total, 24 (24.24%) patients had an adverse event (disease relapse or disease progression), with no significant difference in incidence between children and adolescents. ED patients experienced adverse events in 87.5% of the cases compared to 18.68% in NED patients. The median DFI and EFI were not reached (Figure 1A,C), while the median PFI was 5 years (95%CI > 1.8 years; Figure 1B). The DFI, PFI, and EFI for the entire cohort are depicted in Figure 1D. No significant differences in DFI, PFI, or EFI were observed between the two age groups. No deaths were recorded during the entire follow-up period.

### 3.3. Potential Prognostic Factors for Long-Term Outcomes

The ROC curve analysis (Figure 2) identified a tumor size of 30.5 mm as the most reliable cutoff for predicting adverse events, with a sensitivity of 54.17% (95% CI: 33.33–75.00%), a specificity of 85.33% (95% CI: 76.00–92.00%), and an AUC of 71.14% (95% CI: 58.83–83.44%; *p* = 0.002). Similarly, a total number of positive lymph nodes of 10.5 emerged as the most reliable predictor of adverse event occurrence, with a sensitivity of 83.33% (95% CI: 66.67–95.83%), a specificity of 77.33% (95% CI: 68.00–86.67%), and an AUC of 85.86% (95% CI: 78.21–93.51%; *p* = 3.8 × 10^−7^).

Tumor size greater than 30.5 mm, presence of distant metastases at the diagnosis, capsular invasion, a total number of positive lymph nodes exceeding 10.5, and extrathyroidal tumor extension were identified as potential prognostic factors for the occurrence of adverse events and EFI. The results of the analysis are presented in Table 4.

### 3.4. Univariate and Multivariate Regression Analysis and Risk Stratification

The results of univariate analysis (logistic regression in Table 5; Cox proportional hazards regression in Table 6) confirmed the same prognostic factors for the adverse events and EFI, respectively. However, multivariate analysis (logistic regression in Table 5; Cox proportional hazards regression in Table 6) confirmed only a total number of positive lymph nodes greater than 10.5 and an extrathyroidal tumor extension as significant independent predictors of adverse events and EFI.

Patients were further stratified into risk categories based on the prognostic factors identified through regression analysis:Low-risk (LR) group: patients with a total number of positive lymph nodes fewer than 10.5 and without extrathyroidal extension;Intermediate-risk (IR) group: patients with either a total number of positive lymph nodes exceeding 10.5 or extrathyroidal tumor extension;High-risk (HR) group: patients with both total number of positive lymph nodes exceeding 10.5 and extrathyroidal tumor extension

The EFI for all three risk groups is illustrated in Figure 3, with a statistically significant difference between all three groups (log-rank test: *p* = 1.8 × 10^−10^). The median EFI for the LR and IR groups was not reached, while the median EFI for the HR group was 2.4 years (95% CI: >1.6 years). The ten-year event-free rates were 96% (95% CI: 90.8–100%) for LR patients, 77.1% (95% CI: 61.2–97.2%) for IR patients, and 38.3% (95% CI: 22.5–65.5%) for HR patients.

## 4. Discussion

Risk factors in DTC are usually divided into two groups: patient- (age and sex) and tumor-related (histological type, size, multifocality, extrathyroidal extension, lymph node, and distant metastases). Our study did not identify any patient-related risk factors as being associated with adverse event occurrence or EFI, with similar results being published by other teams [17,18]. Multivariate logistic regression identified a total number of positive lymph nodes exceeding 10.5 and extrathyroidal tumor extension as independent predictors of adverse events. The presence of metastatic disease at initial diagnosis demonstrated a strong trend toward predicting adverse events; however, it did not achieve statistical significance (OR: 9.0, 95% CI: 0.9–93, *p* = 0.06), likely due to the limited sample size. In multivariate Cox regression analysis, the total number of positive lymph nodes greater than 10.5 emerged as the most significant predictor of shorter EFI, with extrathyroidal tumor extension also independently associated with EFI. The patients were later stratified into three risk groups, LR, IR, and HR, according to risk factors detected in regression analysis. Among the 52 patients in the LR group in our study, 25 did not undergo RAI treatment, while 27 did, with no significant difference in the number of adverse events and EFI between these two subgroups. In the LR group, all patients achieved NED status following treatment. Conversely, among the eight patients classified as ED after treatment, six belonged to the HR group, while only two were categorized as IR. The time-to-event analysis demonstrated that patients classified as NED after treatment experienced significantly better long-term outcomes compared to those classified as ED (Figure 1D). The ten-year event-free rate of 96% (95% CI: 90.8–100%) observed in LR patients highlights the excellent treatment outcomes achieved in this group. In contrast, the significantly lower event-free rate of 38.3% (95% CI: 22.5–65.5%) in HR patients underscores the prognostic impact of risk stratification.

Although pediatric and adolescent patients often present with more advanced disease at diagnosis, they generally demonstrate a more favorable prognosis compared to adults [1]. The ATA guidelines for adult patients recommend lobectomy as an appropriate treatment option for individuals classified as LR [19]. However, despite the favorable prognosis associated with DTC in pediatric and adolescent populations, the current ATA guidelines for pediatric patients advocate for total thyroidectomy, regardless of risk level [12]. Nevertheless, evidence from several studies suggests that lobectomy may also represent an appropriate treatment option for LR pediatric patients [20,21,22]. Massimino et al. demonstrated comparable clinical outcomes between radical and conservative approaches, although radical surgeries were associated with significantly higher complication rates [23]. Consistent with these findings, the results of our study highlight the potential for less aggressive treatment strategies in select patient groups. Approaches such as lobectomy and the omission of RAI treatment may represent viable options within this framework.

Identifying patients at elevated risk of disease relapse or progression while carefully selecting those who may benefit from a more conservative treatment approach is a critical component of the treatment process. While numerous studies have attempted to identify potential risk factors, often yielding conflicting results, others have concentrated on improving treatment strategies through the development of risk stratification systems. Over time, several risk stratification systems have been proposed, including TNM/AJCC staging, ATA pediatric risk stratification, and dynamic risk stratification (DRS). Each system offers unique strengths and limitations. The TNM/AJCC staging system, while widely used, is primarily designed to assess mortality risk and has limited discriminatory power in younger populations where overall survival approaches 100% [14]. All patients under 55 years are classified as stage I or II solely based on M status (M0 or M1), which excludes other well-established mortality predictors and reduces its applicability in assessing outcomes in pediatric populations [14]. In contrast, ATA pediatric risk stratification focuses on predicting persistent cervical disease or distant metastases following initial treatment. This system considers factors such as tumor size, extrathyroidal extension, and lymph node metastases but lacks specific thresholds for the number of positive lymph nodes and does not incorporate therapy response, which can significantly influence long-term outcomes [1,12]. DRS addresses some of these gaps by dynamically incorporating therapy response variables. This system allows patients to be reclassified over time into categories such as excellent response, biochemical incomplete response, structural incomplete response, or indeterminate response [24,25]. While DRS has been validated in adult populations, few studies have evaluated its applicability in pediatric DTC [18,26,27]. Given that our study included a patient population treated over a span of 44 years, during which diverse diagnostic tools and laboratory assays were employed for disease evaluation, the full implementation of the DRS system was not feasible. Instead, we proposed a simplified risk stratification system to investigate the potential for adopting a less radical treatment approach. Nevertheless, further multicenter and prospective studies are required to validate our findings and explore their applicability in diverse patient populations and clinical settings.

Additionally, our study showed comparable TNM staging and long-term outcomes, both in the number of adverse events and in the time-to-event analysis, between children and adolescents. The adverse event rate of 24.24% observed in our study was comparable to findings reported in other studies [20,21]. Some studies have suggested that the female-to-male ratio increases with age, particularly after puberty, likely due to hormonal influences [4,28,29,30]. A possible explanation for why the same difference was not observed in our study could lie in the cut-off age of 16 years, which included both pre-pubertal and post-pubertal patients within the children group. Consequently, this age grouping may have made the differences within the children group more pronounced. Multifocality, capsular invasion, and extrathyroidal extension were also more frequent in children, which aligns with findings from other studies [11,31].

This study comes with several limitations. The retrospective nature of the study limits the ability to control for confounding variables and standardize data collection. The tertiary center setting raises the potential for selection bias, which may restrict the generalizability of the findings. Data inconsistencies and incompleteness, such as the inability to distinguish between minimal and gross extrathyroidal extension, stemming from variations in record-keeping and clinical practices over the decades, present additional challenges. The total positive lymph node number cut-off value proposed in this study may not be broadly applicable due to specific IORS’ treatment protocols and given that patients were treated by high-volume specialized endocrine surgeons. Furthermore, the single-center design limits the extrapolation of the results to wider populations.

## 5. Conclusions

This study’s findings further underscore the importance of individualized, risk-based treatment strategies in pediatric DTC. Despite the challenges of managing a cohort treated over a four-decade span with evolving diagnostic and therapeutic approaches, the results highlight the potential for excellent prognosis with tailored management. Further studies are needed to validate these findings, refine risk stratification systems, and establish guidelines for optimizing treatment strategies in this population.

## Figures and Tables

**Figure 1 diagnostics-15-00399-f001:**
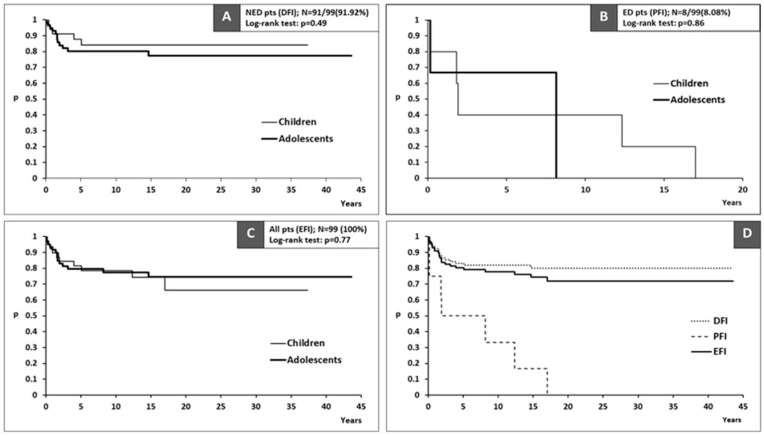
Disease outcomes in the study population: (**A**) DFI for NED patients; (**B**) PFI for ED patients; (**C**) EFI for all patients; (**D**) DFI, PFI, and EFI for all patients. Legend: NED—no evidence of disease; ED—evidence of disease; pts—patients; DFI—disease-free interval; PFI—progression-free interval; EFI—event-free interval.

**Figure 2 diagnostics-15-00399-f002:**
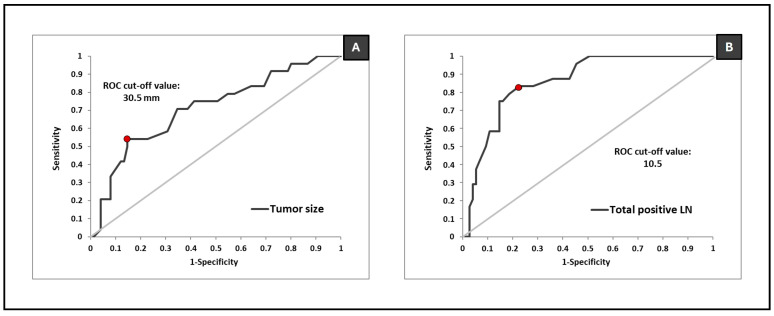
ROC curve analysis cut-off value for predicting adverse events for (**A**) tumor size and (**B**) total number of positive lymph nodes.

**Figure 3 diagnostics-15-00399-f003:**
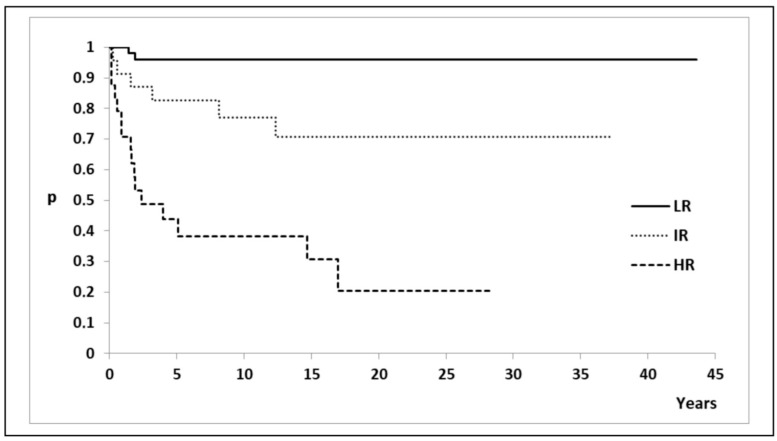
Event-free interval after risk stratification according to regression analysis. Legend: LR—low-risk group; IR—intermediate-risk group; HR—high-risk group.

**Table 1 diagnostics-15-00399-t001:** Demographical and treatment characteristics of the study population.

Characteristics	Total	Children (≤16)	Adolescents (>16)	Pearson χ^2^ Test
**Follow-up (years)**				
Mean (SD)	16.2 (10.3)	15.8 (10.7)	16.5 (10.1)	0.58 ^#^
Median (Range)	15.6 (0.6–43.6)	14.8 (0.7–39.9)	16.3 (0.6–43.6)	
**Patient characteristics**				
** *Sex* **				
Female	75 (75.76%)	29 (74.36%)	46 (76.67%)	0.79
Male	24 (24.24%)	10 (25.64%)	14 (23.33%)	
** *Age (years)* **				
Mean (SD)	16.66 (3.41)	13.08 (2.14)	18.98 (1.61)	-
Median (Range)	17 (7–21)	13 (7–16)	20 (17–21)	
** *Preoperative RT ** **				
Yes	4 (4.04%)	3 (7.69%)	1 (1.67)	0.29 ^♦^
No	95 (95.96%)	36 (92.31%)	59 (98.33%)	
**Surgery**				
** *Thyroid* **				
TT	96 (96.97%)	39 (100%)	57 (95%)	0.27 ^♦^
Near TT	3 (3.03%)	0 (0%)	3 (5%)	
** *CND* **				
Yes	94 (94.95%)	37 (94,87%)	57 (95%)	1.00 ^♦^
No	5 (5.05%)	2 (5.13%)	3 (5%)	
** *SLNB* **				
Yes	50 (50.51%)	18 (46.15%)	32 (53.33%)	0.48
No	49 (49.49%)	21 (53.85%)	28 (46.67%)	
** *MRND* **				
Unilateral	50 (50.51%)	18 (46.15%)	32 (53.33%)	0.10
Bilateral	18 (18.18%)	11 (28.21%)	7 (11.67%)	
No	31 (31.31%)	10 (25.64%)	21 (35%)	
**RAI treatment**				
Yes	74 (74.75%)	32 (82.05%)	42 (70%)	0.17
No	25 (25.25%)	7 (17.95%)	18 (30%)	
**Postoperative external beam RT**				
Yes	4 (4.04%)	1 (2.56%)	3 (5%)	1.00 ^♦^
No	95 (95.96%)	38 (97.44%)	57 (95%)	
**Total**	99 (100%)	39 (100%)	60 (100%)	

SD—standard deviation; RT—radiotherapy; NL—neck lymphadenopathy; TT—total thyroidectomy; CND—central neck dissection; SLNB—sentinel lymph node biopsy; MRND—modified radical neck dissection; RAI—radioactive iodine; ns—no statistical significance; ^#^ Wilcoxon rank sum test; ^♦^ Fischer exact test; * for other primary malignancy.

**Table 2 diagnostics-15-00399-t002:** Pathohistological characteristics and TNM staging of the study population.

Characteristics	Total	Children (≤16)	Adolescents (>16)	Pearson χ^2^ Test (*p*)
**Thyroid cancer type**				
PTC	95 (95.96%)	37 (94.87%)	58 (96.67%)	0.73 ^♦^
FTC	3 (3.03%)	2 (5.13%)	1 (1.67%)	
Hürthle cell	1 (1.01%)	0 (0%)	1 (1.67%)	
**TNM staging ***				
** *pT* **				
pT1a	20 (20.20%)	7 (17.95%)	13 (21.67%)	0.06
pT1b	28 (28.28%)	7 (17.95%)	21 (35%)	
pT2	17 (17.17%)	5 (12.82%)	12 (20%)	
pT3	20 (20.20%)	11 (28.21%)	9 (15%)	
pT4	14 (14.14%)	9 (23%)	5 (8.33%)	
** *pN* **				
pN0	27 (27.27%)	11 (28.21%)	16 (26.67%)	0.72 ^♦^
pN1a	8 (8.08%)	2 (5.13%)	6 (10%)	
pN1b	64 (64.65%)	26 (66.67%)	38 (63.33%)	
** *cM* **				
cM0	91 (91.92%)	34 (87.18%)	57 (95%)	0.25 ^♦^
cM1	8 (8.08%)	5 (12.82%)	3 (5%)	
**Tumor size (mm)**				
Mean (SD)	22.3 (14.94)	26.23 (16.77)	19.75 (13.14)	0.06 ^#^
Median (range)	19 (0–70)	22 (1–70)	18 (0–60)	
**Central lymph nodes (number)**				
** *Positive* **				
Mean (SD)	5.1 (6)	5.23 (5.57)	5.02 (6.3)	0.63 ^#^
Median (range)	4 (0–29)	4 (0–20)	3 (0–29)	
** *Total* **				
Mean (SD)	10. 76 (7.66)	10.82 (7.2)	10.72 (8)	0.80 ^#^
Median (range)	8 (0–30)	11 (0–25)	7.5 (0–30)	
**Lateral lymph nodes (number)**				
** *Positive* **				
Mean (SD)	4.69 (6.18)	6.26 (7.04)	3.67 (5.36)	0.10 ^#^
Median (range)	2 (0–25)	4 (0–24)	2 (0–25)	
** *Total* **				
Mean (SD)	17.45 (13.4)	18.82 (14.84)	16.57 (12.42)	0.60 ^#^
Median (range)	16 (0–60)	18 (0–60)	12 (0–48)	
**Multifocality**				
No	50 (50.51%)	12 (30.77%)	38 (63.33%)	**0.001**
Yes	49 (49.49%)	27 (69.23%)	22 (36.67%)	
**Capsular invasion**				
No	44 (44.44%)	10 (25.64%)	34 (56.67%)	**0.002**
Yes	55 (55.56%)	29 (74.36%)	26 (43.33%)	
**Extrathyroidal extension (minimal and gross)**				
No	65 (65.66%)	21 (53.85%)	44 (73.33%)	**0.046**
Yes	34 (34.34%)	18 (46.15%)	16 (26.67%)	
**Evidence of disease**				
NED	91 (91.92%)	34 (87.18%)	57 (95%)	0.25 ^♦^
ED	8 (8.08%)	5 (12.82%)	3 (5%)	
**Total**	99 (100%)	39 (100%)	60 (100%)	

PTC—papillary thyroid cancer; FTC—follicular thyroid cancer; TNM—tumor, node, metastasis; * AJCC/UICC 8th edition; SD—standard deviation; Extrathyroidal extension (minimal and gross)—spread of thyroid cancer beyond the thyroid gland into surrounding tissues; NED—no evidence of disease; ED—evidence of disease; ns—no statistical significance; ^#^ Wilcoxon rank sum test with continuity correction; ^♦^ Fischer exact test.

**Table 3 diagnostics-15-00399-t003:** Disease outcomes in the study population.

Outcome	Total	Children (≤16)	Adolescents (>16)	Pearson χ^2^ Test
**Event**				
** *NED pts* **				
With recurrence	17 (18.68%)	5 (14.71%)	12 (21.05%)	0.45
Without recurrence	74 (81.32%)	29 (85.29%)	45 (78.95%)	
Total	91 (100%)	34 (100%)	57 (100%)	-
** *ED pts* **				
With progression	7 (87.5%)	5 (100%)	2 (66.67%)	0.37 ^♦^
Without progression	1 (12.5%)	0 (0%)	1 (33.33%)	
Total	8 (100%)	5 (100%)	3 (100%)	-
** *Adverse events in all pts* **				
With adverse event	24 (24.24%)	10 (25.64%)	14 (23.33%)	0.79
Without adverse event	75 (75.76%)	29 (74.36%)	46 (76.67%)	
Total	99 (100%)	39 (100%)	60 (100%)	-
**Time to event**				
** *DFI (years) for NED pts* **				
Median (95%CI)	NR	NR	NR	0.49 *
** *PFI (years) for ED pts* **				
Median (95%CI)	5 (>1.8)	1.9 (>1.8)	8.1 (>0.2)	0.86 *
** *EFI (years) for all pts* **				
Median (95%CI)	NR	NR	NR	0.77 *

NED—no evidence of disease; pts—patients; ED—evidence of disease; DFI—disease-free interval; PFI—progression-free interval; EFI—event-free interval; CI—confidence interval; NR—not reached; ^♦^ Fischer exact test; * Log-rank test.

**Table 4 diagnostics-15-00399-t004:** Demographic and pathological characteristics as prognostic factors.

Characteristics	Event (Relapse/Progression)			Event-Free Interval (Years)	
	Without	With	Test	Median EFI (95%CI)	Log-Rank Test
** *Age (years)* **					
Children	29 (38.67%)	10 (41.67%)	0.79	NR (>17)	0.77
Adolescents	46 (61.33%)	14 (58.33%)		NR	
** *Sex* **					
Female	56 (74.67%)	19 (79.17%)	0.65	NR	0.73
Male	19 (25.33%)	5 (20.83%)		NR	
** *Tumor size (ROC)* **					
≤30.5 mm	64 (85.33%)	11 (45.83%)	**8.5 × 10^−5^**	NR	**3.2 × 10^−6^**
>30.5 mm	11 (14.67%)	13 (54.17%)		4.8 (>1.6)	
** *Multifocality* **					
No	41 (54.67%)	9 (37.5%)	0.14	NR	0.17
Yes	34 (45.33%)	15 (62.5%)		NR (>17)	
** *cM1 at diagnosis* **					
No	74 (98.67%)	17 (70.83%)	**1.5 × 10^−4^**	NR	**1.7 × 10^−6^**
Yes	1 (1.33%)	7 (29.17%)		5 (>1.8)	
** *Capsular invasion* **					
No	41 (54.67%)	3 (12.5%)	**2.9 × 10^−4^**	NR	**3.3 × 10^−4^**
Yes	34 (45.33%)	21 (87.5%)		NR (>12.3)	
** *Positive LN (ROC)* **					
≤10.5	58 (77.33%)	4 (16.67%)	**1.5 × 10^−7^**	NR	**2.7 × 10^−8^**
>10.5	17 (22.67%)	20 (83.33%)		12.3 (>3.2)	
** *Extrathyroidal extension (minimal and gross)* **					
No	59 (78.67%)	6 (25%)	**1.4 × 10^−6^**	NR	**2 × 10^−7^**
Yes	16 (21.33%)	18 (75%)		14.7 (>1.8)	
**Total**	75 (100%)	24 (100%)	-	-	-

ROC—receiver operating characteristic curve analysis cut-off value; cM1—clinically verified distant metastasis, LN—lymph node; Extrathyroidal extension (minimal and gross)—spread of thyroid cancer beyond the thyroid gland into surrounding tissues; EFI—event-free interval; CI—confidence interval; NR—not reached; ns—not statistically significant.

**Table 5 diagnostics-15-00399-t005:** Univariate and multivariate logistic regression analysis results for adverse event occurrence.

Characteristics	Logistic Regression for Event (Relapse/Progression)			
	Univariate		Multivariate	
	OR (95%CI)	Wald Test	OR (95%CI)	Wald Test
** *Tumor size (ROC)* **				
>30.5 mm:≤30.5 mm	6.9 (2.5–19.2)	**2.3 × 10^−4^**	-	0.09
** *Initially metastatic* **				
Yes:No	30.5 (3.5–264.2)	**1.9 × 10^−3^**	-	0.06
** *Capsular invasion* **				
Yes:No	8.4 (2.3–30.7)	**1.2 × 10^−3^**	-	0.81
** *Positive LN (ROC)* **				
>10.5:≤10.5	17.1 (5.1–56.7)	**3.7 × 10^−6^**	6.4 (1.7–24.6)	**7 × 10^−3^**
** *Extrathyroidal extension (minimal and gross)* **				
Yes:No	11.1 (3.8–32.5)	**1.2 × 10^−5^**	5.1 (1.4–18.3)	**0.01**

ROC—receiver operating characteristic curve analysis cut-off value; LN—lymph node; Extrathyroidal extension (minimal and gross)—the spread of thyroid cancer beyond the thyroid gland into surrounding tissues; OR—odds ratio; CI—confidence interval; ns—not statistically significant.

**Table 6 diagnostics-15-00399-t006:** Univariate and multivariate Cox proportional hazard regression analysis results for the event-free interval.

Characteristics	Cox Regression for EFI			
	Univariate		Multivariate	
	HR (95%CI)	Wald Test	HR (95%CI)	Wald Test
** *Tumor size (ROC)* **				
>30.5 mm:≤30.5 mm	5.63 (2.49–12.71)	**3.2 × 10^−5^**	-	0.16
** *Initially metastatic* **				
Yes:No	6.6 (2.72–16.01)	**3 × 10^−5^**	-	0.14
** *Capsular invasion* **				
Yes:No	6.76 (2.01–22.69)	**1.9 × 10^−3^**	-	0.84
** *Positive LN (ROC)* **				
>10.5:≤10.5	11.36 (3.87–33.36)	**9.8 × 10^−6^**	6.48 (2.05–20.51)	**2.1 × 10^−6^ ***
** *Extrathyroidal extension (minimal and gross)* **				
Yes:No	8.02 (3.17–20.29)	**1.1 × 10^−5^**	3.83 (1.42–10.36)	

ROC—receiver operating characteristic curve analysis cut-off value; LN—lymph node; Extrathyroidal extension (minimal and gross)—the spread of thyroid cancer beyond the thyroid gland into surrounding tissues; EFI—event-free interval; HR—hazard ratio; CI—confidence interval; ns—not statistically significant; * Likelihood ratio test.

## Data Availability

To protect the privacy and confidentiality of patients in this study, clinical data have not been made publicly available in a repository of the article; however, they will be made available upon reasonable request to the corresponding author.

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
