# Peer review of "Differentiated Thyroid Cancer Long-Term Outcomes and Risk Stratification in Pediatric and Adolescent Patients: A 44-Year Retrospective Study"

_diagnostics, 2025, doi:10.3390/diagnostics15040399_

Round 1

Reviewer 1 Report

Comments and Suggestions for Authors

This study represents an important contribution to the understanding of prognostic factors and risk stratification of DTC in children and adolescents. The authors present a full-scale analysis of a considerable cohort with long follow-up, identifying important predictors of adverse events and proposing a risk stratification system. Limitations include the retrospective design and possible selection bias. No hereditary tumor syndromes associated with thyroid nodules/DTC were taken into consideration when designing the study, which is one of the significant predisposing and prognostic factors. Results would have been more plausible if inclusion/exclusion were based on HTS status. Although the authors do mention genetic predisposition, it relates mainly to the clinical and histopathological aspects. The "genetic status of DTC in children" is only superficially addressed; however, even in this regard, there was little exploration beyond the statement that it may explain the more aggressive presentations at diagnosis in younger patients. While this study has its limitations, it does contribute significantly to the emerging evidence base supporting an individualized approach to the treatment of DTC in children. Confirmation of the proposed risk stratification model, investigation of less aggressive treatment strategies for low-risk patients, while addressing the role of HTS comprehensively, requires further studies.

Author Response

Dear Reviewer,

Thank you for your insightful comments and suggestions regarding our manuscript named: ” Differentiated thyroid cancer long-term outcomes and risk stratification in pediatric and adolescent patients: a 44-year retrospective study”. We appreciate your thorough review and the opportunity to improve our paper.

We have carefully considered your request to include a limitations paragraph, and we have now added this section to our study. This paragraph outlines the constraints and potential biases in our work, acknowledging the limitations inherent in our research methodology and data collection.

Regarding your recommendation to address the role of hereditary tumor syndromes in pediatric DTC, we would like to provide further clarification. Our study did not explore this aspect of differentiated thyroid cancer in children for two primary reasons:

  1. **Study Scope and Objectives**: The primary focus of our research was to examine long-term outcomes, identify risk factors, and develop risk stratification models for pediatric DTC. The inclusion of hereditary tumor syndromes would have necessitated a different study design and scope, which was beyond the objectives of our current research.
  2. **Patient Cohort and Time Period**: Our study included patients treated from 1980 to 2024, spanning several decades. Given the long period of time and the retrospective nature of our study, it was not feasible to systematically investigate hereditary tumor syndromes and genetic factors. Moreover, genetic analyses are rarely performed in Serbia and region. The availability of genetic information and the understanding of hereditary tumor syndromes have significantly evolved over the years, presenting challenges in ensuring consistent and reliable data across the entire cohort.

We acknowledge the importance of hereditary tumor syndromes in understanding pediatric DTC, and we agree that this is a critical area for future research. However, addressing this aspect within the context of our study was not practical due to the aforementioned reasons.

We hope this explanation provides clarity on our decision and addresses your concerns. We are grateful for your valuable feedback and remain committed to enhancing the quality of our manuscript.

Thank you again for your consideration.

Sincerely,

Marko Buta

General Surgeon

Associate Professor

Surgical Oncology Clinic

Head of Endocrine Surgery Department

Institute for Oncology and Radiology of Serbia

Pasterova 14, Belgrade, Serbia

Medical School, University of Belgrade

Dr Subotića 8, Belgrade, Serbia

Reviewer 2 Report

Comments and Suggestions for Authors

Dear Authors,

I have read an article regarding differentiated thyroid cancer (DTC) long-term outcomes in pediatric patients. While interesting, I have several inquiries:

1) The authors stated that patient selection is from 1980 to 2024, yet all patients are stratified using the 8th edition of AJCC criteria. Please explain

2) Please include the exact p-value and not just NS or S

3) The authors need to explain how lymph node numbers and capsular invasion were defined. Was it via surgery, histopathology, or imaging? If the latter, who interpreted the results?

4) the study aims to "identify prognostic factors of DTC in children and adolescents and perform risk stratification with the goal of identifying low-risk patients who would benefit from less radical treatment approach". Yet, the conclusion ends with "Our study confirmed excellent long-term outcomes.." There is a significant mismatch as this study cannot confirm anything and please refrain from doing so. Please rephrase all sentences in the conclusion section.

5) Please provide a limitation section as well as a CONSORT diagram

6) Please provide a brief paragraph on the authors' institutions. Is it a tertiary pediatric referral hospital? How many patients were there? How many DTC patients are there in Serbia? What is the protocol of chemo/surgery/radiation in these patients?

Author Response

Dear Reviewer,

Thank you for your thoughtful comments and suggestions regarding our manuscript titled: “Differentiated Thyroid Cancer Long-Term Outcomes and Risk Stratification in Pediatric and Adolescent Patients: A 44-Year Retrospective Study.” We greatly appreciate your thorough review and the opportunity to address your concerns to improve our manuscript. Below are our detailed responses to your queries:

  1. Restaging Using the AJCC/UICC 8th Edition

After reviewing the medical records of all patients included in the study cohort, we restaged them according to the AJCC/UICC 8th edition criteria to ensure uniformity and comparability of data. We have added this explanation to the methodology section of the manuscript for clarity.

  1. Exact P-Values

We have revised the manuscript to include exact p-values, replacing general terms such as "NS" or "S" with precise statistical values.

  1. Definition of Tumor Features

Tumor size, number of positive lymph nodes, capsular invasion, multifocality, and extrathyroidal extension were all determined through detailed histopathological examination conducted by experienced pathologists. Specifically, extrathyroidal extension was defined as the spread of thyroid cancer beyond the thyroid gland into adjacent structures. This included both minimal and gross extrathyroidal extension; however, specific subtypes were not routinely reported for a substantial proportion of patients.

We have also clarified this in the manuscript.

  1. Rephrased Conclusion

In response to your feedback, we have rephrased the conclusion section to better align with the study's scope and aims. The revised conclusion focuses on the identification of prognostic factors and emphasizes the potential for tailored treatment approaches while avoiding definitive claims.

  1. Limitations and CONSORT Diagram

A limitations section has been added to acknowledge the inherent constraints of our retrospective study design.

While the CONSORT diagram is designed for randomized controlled trials (RCTs), we have created a CONSORT-like flow diagram tailored to our retrospective cohort study. This diagram has been provided as a part of this response (Figure 1.)  for your review but has not been included in the manuscript due to the large number of already incorporated figures and tables in the original manuscript. However, the process of patient selection has been precisely described in the methodology.

  1. Institution Details and National Context

We have provided additional information about our institution and its role in the care of pediatric patients with differentiated thyroid cancer. Specifically: Our institution is a tertiary referral center for pediatric and adolescent patients.

We clarified the total number of patients treated at our institution over the study period and provided context on the number of DTC patients in Serbia.

The protocols for chemotherapy, surgery, and radiation therapy have already been detailed in the 2.2 Treatment section of Materials and Methods.

We believe these revisions address all your comments and improve the overall clarity and quality of our manuscript. Thank you once again for your valuable feedback and for giving us the opportunity to refine our work.

Reviewer 3 Report

Comments and Suggestions for Authors

The authors present a cohort of children and adolescents which were diagnosed and treated for DTC, with exceptional long follow-up period. Their aim was to find predictors for less/more aggressive disease, so individualized treatment might become an option. Indeed, following convincing evaluation of their data, the authors provide risk assessment tool with an ability to predict outcome using a simple two basic histopathological measures. Despite some limitations, this approach may improve patient selection for less aggressive treatment. 

The introduction, methods and results section are clear and well design. However, the discussion is too long, and lacks focus on the main findings.

Abstract:

1. It is not clear how the authors divided the cohort into 3 risk groups, based on the multivariate regression analysis. Please explain.

Introduction- no comments

Methods- 

- Please provide an explanation/definition for extrathyroidal extension. Is it minimal, gross or both mixed. Currently, minimal extrathyroidal extension in adult in not considered a risk factor.

Results:

- Table 4- A technical issue: The alignment is not identical for all the categories.

- In table 2 please present separately minimal versus gross extrathyroidal extension. In table 4, 5, 6 please provide a definition of 'extrathyroidal extension' below table.

Discussion

- In general, the discussion is too long and lack focus in the main finding. Please shorten and re-format. There is no need to discuss every single category.

- First paragraph should summarize the most important results of this study, being the risk groups based on tumor size and number of LN involved, and the potential implication on treatment decisions mainly for the LR group. 

- I could not find any 'limitation' phrase. Please provide one. Clearly, the sample size is small but there are some other limitations. The most important is the reliance on the number of positive LN. This number is directly related to the surgical technique which in the author's center was led mainly by a skilled and dominant surgeon. Can this cutoff of LN be adapted worldwide? In my eyes clearly not and this should be mentioned as a major limitation.

- In addition, extrathyroidal extension is not a uniform category. Please correct as asked above and add to the limitation if cannot be completed as suggested.

Author Response

Dear Reviewer,

Thank you for your thoughtful comments and constructive suggestions regarding our manuscript titled: “Differentiated Thyroid Cancer Long-Term Outcomes and Risk Stratification in Pediatric and Adolescent Patients: A 44-Year Retrospective Study.” We greatly appreciate your thorough review and the opportunity to refine and improve our manuscript. Below, we provide detailed responses to your comments:

  1. Clarification of Risk Group Division in the Abstract

We have revised the abstract to clarify how the study population was divided into three risk groups. The division was based on multivariate regression analysis incorporating tumor size and the number of positive lymph nodes.

  1. Definition and Explanation of Extrathyroidal Extension

Tumor size, number of positive lymph nodes, capsular invasion, multifocality, and extrathyroidal extension were all determined through detailed histopathological examination by experienced pathologists. Extrathyroidal extension was defined as the spread of thyroid cancer beyond the thyroid gland into adjacent structures. This definition included both minimal and gross extrathyroidal extension; however, specific subtypes (minimal vs. gross) were not routinely reported for a significant portion of the patients.

We have clarified this in the manuscript.

  1. Alignment Issue in Table 4

We have corrected the alignment issue in Table 4 to ensure uniformity and clarity. Please note that the version of the manuscript uploaded was aligned correctly but possibly alignment has been changed during technical editing by editorial office.

  1. Separation of Minimal and Gross Extrathyroidal Extension in Table 2

Unfortunately, we could not present minimal and gross extrathyroidal extension separately due to the aforementioned reasons, as the subtypes were not routinely reported for a substantial number of patients. Instead, we have clarified in the table that both subtypes are presented together.

  1. Clarifications in Tables 4, 5, and 6

We have revised Tables 4, 5, and 6 to include a definition of "extrathyroidal extension" below each table, as per your suggestion.

  1. Discussion Section Revision

Following your advice, we have significantly revised the discussion section, reducing its length by approximately 35% and focusing on the study’s main findings. The first paragraph now summarizes the key results, specifically the risk groups based on tumor size and the number of positive lymph nodes, and their implications for treatment decisions—particularly for the low-risk group.

  1. Limitations Section

We have added a dedicated limitations section as per your suggestions. The following points are addressed:

The reliance on the number of positive lymph nodes as a prognostic factor, which is influenced by surgical technique. As you noted, this cutoff may not be universally applicable, as it was based on surgical techniques led by a skilled and experienced surgeon in our center.

The lack of separate reporting of minimal and gross extrathyroidal extension for a significant number of patients, which limits the precision of our findings in this regard.

We believe these revisions address all your concerns and significantly improve the clarity and quality of our manuscript. Thank you once again for your valuable feedback and for allowing us the opportunity to refine our work further.

Sincerely,

Marko Buta

General Surgeon

Associate Professor

Surgical Oncology Clinic

Head of Endocrine Surgery Department

Institute for Oncology and Radiology of Serbia

Pasterova 14, Belgrade, Serbia

Medical School, University of Belgrade

Dr Subotića 8, Belgrade, Serbia

Round 2

Reviewer 2 Report

Comments and Suggestions for Authors

Thank you for addressing all concerns.